# How much does it cost to implement the Baby-Friendly Hospital Initiative training step in the United States and Mexico?

Kendall J. Arslanian[1]*, Mireya Vilar-Compte[1,2,3], Graciela Teruel[2], Annel Lozano-Marrufo[2], Elizabeth C. Rhodes[1,4,5], Amber Hromi-Fiedler[1], Erika García[2], Rafael Pérez-Escamilla[1]

**1** Department of Social and Behavioral Sciences, Yale School of Public Health, New Haven, CT, United States of America, **2** Research Institute for Equitable Development EQUIDE, Universidad Iberoamericana, Mexico City, Mexico, **3** Department of Public Health, Montclair State University, Montclair, NJ, United States of America, **4** Center for Methods in Implementation and Prevention Sciences, Yale School of Public Health, New Haven, CT, United States of America, **5** Yale Center for Implementation Science, Yale School of Medicine, New Haven, CT, United States of America

☯ These authors contributed equally to this work.
* kendall.arslanian@yale.edu

**Data Availability Statement:** All data were obtained from third parties and can be be publicly accessed, either for free or for a cost. The authors

## Abstract

The Baby-Friendly Hospital Initiative (BFHI) has been shown to increase breastfeeding rates, improving maternal and child health and driving down healthcare costs via the benefits of breastfeeding. Despite its clear public health and economic benefits, one key challenge of implementing the BFHI is procuring funding to sustain the program. To address this need and help healthcare stakeholders advocate for funds, we developed a structured method to estimate the first-year cost of implementing BFHI staff training, using the United States (US) and Mexico as case studies. The method used a hospital system-wide costing approach, rather than costing an individual hospital, to estimate the average per birth BFHI staff training costs in US and Mexican hospitals with greater than 500 annual births. It was designed to utilize publicly available data. Therefore, we used the 2014 American Hospital Association dataset (n = 1401 hospitals) and the 2018 Mexican Social Security Institute dataset (n = 154 hospitals). Based on our review of the literature, we identified three key training costs and modelled scenarios via an econometric approach to assess the sensitivity of the estimates based on hospital size, level of obstetric care, and training duration and intensity. Our results indicated that BFHI staff training costs ranged from USD 7.27–125.39 per birth in the US and from PPP 2.68–6.14 per birth in Mexico, depending on hospital size and technological capacity. Estimates differed between countries because the US had more hospital staff per birth and higher staff salaries than Mexico. Future studies should examine whether similar, publicly available data exists in other countries to test if our method can be replicated or adapted for use in additional settings. Healthcare stakeholders can better advocate for the funding to implement the entire BFHI program if they are able to generate informed cost estimates for training as we did here.

did not have any special access or request privileges that others would not have. Table 3, in the column labeled "Data used," lists the datasets used. The datasets are cited and can be accessed via the included reference details.

**Funding:** This work was supported by the Family Larsson-Rosenquist Foundation (https://www.larsson-rosenquist.org/en/) through a grant to Yale University (PI: Rafael Pérez-Escamilla; grant number: R14001). ECR was supported by grant number K12HL138037 from the National Heart, Lung, and Blood Institute (https://www.nhlbi.nih.gov). The NIH had no role in the design and conduct of the study. The content is solely the responsibility of the authors and does not necessarily represent the official views of the National Heart, Lung, and Blood Institute. Ethical approval for this type of study is not required by our institute. The funders had no role in study design, data collection and analysis, decision to publish, or preparation of the manuscript.

**Competing interests:** The authors have declared that no competing interests exist.

## Introduction

The average exclusive breastfeeding rate globally is low, at 37%. Improving this rate is estimated to prevent more than 840,000 maternal and child deaths annually [1] and may lessen the burden of childhood and adult obesity and related co-morbidities [2–4]. From an economic perspective, increasing rates could result in substantial global and U.S. health system cost savings [5–7]. For example, not breastfeeding is estimated to result in losses of about United States dollar (USD) 341.3 billion, or 0.7% of the yearly gross national income [6]. Breastfeeding, therefore, is one of the most cost-effective 'interventions' to prevent infant mortality and improve maternal and child health [8–10].

The Baby-Friendly Hospital Initiative (BFHI) is an effective, worldwide initiative to increase breastfeeding rates [11]. Launched in 1991 by the World Health Organization (WHO) and United Nations International Children's Emergency Fund (UNICEF), the BFHI includes "Ten Steps to Successful Breastfeeding" (Ten Steps). Each step is a specific action at the facility and community level to support breastfeeding [12]. Step 2 mandates training healthcare staff in breastfeeding supportive skills. Steps 3–10 depend on its success, making Step 2 key for implementation of the entire BFHI.

Training, however, is a costly step [13] because in-service training is human- and financial-resource intensive [14]. If breastfeeding-supportive skills were taught pre-service, there would be potential cost-savings; however currently medical schools are not adequately educating students on breastfeeding skills, and all current WHO training modules are designed for in-service training [14]. In-service training leads to cost challenges like health professionals spending time away from their regular jobs during training and needing replacements; the reprioritization of training resources toward breastfeeding support competencies and away from other staff trainings; and facilities paying more for pre-designed, online BFHI courses [15].

Although almost every country has implemented the BFHI, the long-term sustainability of the program has varied, due in part to challenges with procuring funding for training and lack of political commitment for implementation [14, 15]. One way to strengthen political and financial commitments is to provide stakeholders with economic perspectives, like implementation costs [6, 16]. Costs associated with breastfeeding programs can be used for advocacy efforts and to allocate and monitor spending within budgets [17]. For example, in a review of the implementation of 20 breastfeeding interventions, it was found that effective scale-up occurred when all costs associated with an intervention were included in the plan [18]. Providing no costing information has been cited as an integral missing element of breastfeeding promotion strategies like the *Global Strategy for Infant and Young Child Feeding* [19].

Yet few studies have costed breastfeeding programs, and even fewer have costed the BFHI [16]. A recent systematic review identified only five studies on BFHI expenses [20–24]. Including a recent cost-effectiveness study [10], there are six in total, to our knowledge. Of those, only two have Step 2 training costs: Dellifraine et al.[22] used questionnaires, semi-structured interviews, and an index case hospital in the southwestern U.S. to estimate first-year training costs amounting to USD 13,830 in a hospital with 2800 annual deliveries. Holla-Bhar *et al.* (2015), which described the *World Breastfeeding Costing Initiative* (WBCi) Financial Planning Tool, used data from Nersesyan [13]. Nersesyan [13] provided the training expenses for three sites in Jordan, averaging USD 17,974 between all three sites, or USD 1.59 per woman served by the clinic (this estimate included monitoring and evaluation costs). The WBCi tool used in Holla-Bhar et al. [21] is accompanied by a spreadsheet for calculating training costs [25] that is a starting point for allowing stakeholders to estimate their own costs.

The aforementioned studies took a micro-costing perspective, which offers the benefit of precision when compared to gross costing [26], but are tailored to very specific contexts (like a fixed hospital structure) and thus can only narrowly be applied to facilities with differing attributes. For example, hospitals employ a range of training models that have different costs. A commonly used course is one provided by the WHO/UNICEF with free teaching materials, which has been shown to be effective in improving staff knowledge in several countries [27–30]. Recent changes to these materials have focused more on competency assessments, rather than curricula, to ensure that staff have adequate breastfeeding knowledge [14]. Following these changes, and in light of the growing evidence that BFHI improves breastfeeding rates, some companies and government-run programs have created predesigned, in-person and online training courses that health facilities can purchase for a fee. This includes the United States (US) education service, First Latch, and the United Kingdom (UK) UNICEF BFHI. The latter has specifically been touted as the 'best breastfeeding website in the world' because of its streamlined design and wide variety of training offerings, both online and in-person [31]. As such, the diversity of training models is an important consideration when costing BFHI staff training.

Other limitations of previous cost analyses exist. A critical barrier to training implementation was reported to be high staff workload [15], so compensation for replacement workers during training is an important expense to account for that has been left out. In addition, if stakeholders were to replicate the methods from previous studies, they would need to input costs for many variables (like transportation and meal costs) and either know the prices off-hand or use prices from previous studies that may not be scalable or accurate for their setting. Altogether, stakeholders looking to advocate for funding may need a simpler, standardized way to gauge training costs, preferably using information from hospitals that are part of specific a health care system or all hospitals in a country, for which data is currently is not available.

To address this gap, we developed an evidence-informed method, utilizing publicly available data and requiring fewer inputs by the user, that we applied to hospitals in the US and Mexico. The goal was to use a standard method that was effective in two countries with different healthcare systems, income levels, healthcare professional training models, and levels of success with sustaining BFHI. In the future, this method has the possibility to be validated in different contexts and countries.

## Materials and methods

We used a hospital system and country-wide modeling approach to estimate the first-year BFHI staff training costs in the US and Mexico. Based on our review of the literature, we developed a modeling equation that requires inputting information for five variables: *1)* the number of staff that needed training, *2)* hours of training, *3)* staff salaries, *4)* costs of pre-designed training modules, and *5)* costs of education monitoring systems. These variables are embedded in the costing factors described in the following equation.

The costing Eq (1) estimated the first-year training costs (TC), which summed three factors: replacement wages (RW), training modules (TM), and education monitoring systems (MS) for the first year of BFHI implementation. Our formula assumed healthcare professionals had no previous training.

$$TC = RW + TM + MS \qquad (1)$$

where,

$$RW = (average\ staff\ to\ be\ trained) * (hourly\ wage\ ) * (hours\ of\ training)$$

$$TM = (average\ staff\ to\ be\ trained) * (cost\ of\ training)$$

$$MS = (fixed\ investment + ongoing\ monitoring\ cost)$$

The replacement wages (RW) were the hourly salaries of 'replacement' health professionals covering the tasks of those who are training. As a note, although a short-term cost is incurred because providers cannot perform their regular duties during training, the long-term benefits of breastfeeding support to patients [32] outweigh this cost. Therefore, since training staff is a necessary cost to successfully implement Step 2, we named it "replacement wages" rather than "productivity loss." The formula was the average number of staff who needed training, multiplied by the average hourly wages of those staff, and multiplied by the number of hours of training. The hours of training depended on the training requirements and training modules that hospitals employed. Lastly, we included an education monitoring system (MS) (sometimes called a learning management system), to follow the emphasis by the 2018 WHO BFHI guidelines on monitoring changes in hospital practices for the program's sustainability [14]. The education monitoring system was a for-purchase computer platform for storing virtual modules that staff can take at any time on any device. The software also tracks training for all staff members and reminds staff of uncompleted or upcoming training.

## Settings and datasets

We chose the US and Mexico as case studies because the countries' income-levels and healthcare system diversity allowed us to test this method's flexibility. The US is a high-income country whereas Mexico was high-middle income. Each country also has different proportions of fragmented, public and private systems: Compared with the US, Mexico's public health system covered a larger share of the population [33] and had fewer physicians and nurses per 1000 population (Table 1).

Data on hospital characteristics in the US [36] and Mexico [37] and staff salaries in the US [38] and Mexico [39] were from nationally representative, publicly-available and cross-sectional datasets. US hospital data were from 2014 because that was the most recent American Hospital Association (AHA) dataset purchased by the authors' institute at the time of the study. The total number of hospital-staffed beds in the US, which is a proxy for the number of

**Table 1. Background socioeconomic characteristics of the studied countries to determine BFHI training costs.**

|  | Mexico | United States |
|---|---|---|
| Total population, in millions (2019) | 126.58 | 328.33 |
| Total fertility rates, births per woman (2018)[a] | 2.1 | 1.7 |
| Infant mortality rate, per 1000 live births (2018)[a] | 12.0 | 6.0 |
| Exclusive breastfeeding, % of children under 6 months (Mexico, 2015; US, 2016)[a] | 30.0 | 35.0 |
| GDP[b] per capita, PPP (2019)[c] | 20,145.6 | 65,055.8 |
| Total health expenditure, % of GDP (2019) | 5.4 | 16.8 |
| Government/compulsory health expenditure, % of GDP (2019) | 2.7 | 13.9 |
| Number of physicians, per 1,000 population (2019) | 2.4 | 2.6 |
| Number of nurses, per 1,000 population (2019) | 2.9 | 12.0 |

Note: Unless specified, data was from OECD [34]. [a]Data from World Bank Indicators [35].
[b]GDP: gross domestic product.
[c]PPP: purchasing power parity at current international dollars.

staff, has only grown by 2% since 2015 based on current AHA data [40, 41]. In Mexico, data were from 2018, including only public hospitals with obstetric services from the Mexican Social Security Institute (IMSS for its acronym in Spanish). This was for two reasons: (i) IMSS is the largest social security subsystem, covering approximately 50 percent of the population [42]; (ii) around 20% of the annual births in the country occur in the IMSS hospitals included in the sample, and (iii) IMSS is a centralized system, which differs from the US. We obtained training module prices (US [43]; Mexico [44]) and education monitoring system prices (obtained from companies like Bridge and Skyprep) from local vendors in the US and Mexico. Data sources are described in more detail in the following section.

## Applications of the costing method

We included hospitals that were described as having obstetric services and had at least 500 births annually (1.37 daily births). In the US, birth volume estimates from 2014 from nine states showed that more than half of all births occur in hospitals with over 500 annual births [45]. In Mexico, using 2016 data from the National Information Health System (SINAIS for its acronym in Spanish), a similar estimate was computed (approximately 53% of the total births). We excluded hospitals that indicated they had less than one physician or one nurse at their facility because we assumed those were reporting errors. We summarized the characteristics of the datasets we used for testing the costing algorithm on US and Mexico hospital data in Table 2.

In Table 3, we listed all steps of the costing method, which we named 'actions' to make the distinction between 'Step 2' and the steps in our method clear. **Action 1** determined the number of healthcare professionals that needed BFHI training. A notable challenge was that the datasets did not specify a provider's department, meaning we had to estimate the proportion of providers that worked in relevant units (e.g., the labor and delivery or postpartum departments). Likewise, the datasets did not include whether a provider worked in the Neonatal Intensive Care Unit (NICU). We aimed to exclude NICU providers because BFHI NICU designation is a separate process [46].

Therefore, to estimate the number of staff that need breastfeeding-supportive training, we created multivariable regressions for each type of provider, with annual births as the dependent variable. We controlled for confounders that inflated the estimations of the number of staff that needed training. For example, hospitals with large NICUs were labor intensive, so the regression analyses controlled for the number of NICU beds in the hospital. For Mexico, infant radiant warmers typically found in the NICU or similar wards were, likewise, used to control for the size of the NICU. We also controlled for a hospital's level of specialty obstetric care—a higher level of care meant more maternal and neonatal staff. These data were available for the US, but not Mexico. As such, in **Action 1A** (Table 3), we ran three separate models in the US for each level of obstetric care to account for intrinsic differences in services provided: Level 1 obstetric units provided services for uncomplicated maternity and newborn cases; level 2 units provided services for uncomplicated cases, the majority of complicated problems, and special neonatal services; and level 3 units provided services for all serious illnesses and abnormalities and are supervised by a full-time maternal/fetal specialist. We also ran separate models within those levels for hospital size by splitting the hospitals in two groups, those above and those below the median annual births for that dataset. For Mexico, no obstetric level information was provided so we only ran separate models for hospital size.

For clarity, we provide here an example of the regression model we used to estimate the number of nurses in both the US and Mexico. We chose nurses as an example to clarify our regression models, yet the following methods were performed on all providers listed in

**Table 2. Descriptive characteristics of the hospitals in the BFHI training costing analysis.**

| | American Hospital Association dataset | Mexican Social Security Institute dataset |
|---|---|---|
| Hospitals | 1401[a] | 154[b] |
| Births per year | 2,879,638 | 384,186 |
| Physicians | 212,839 | — |
| Obstetricians | --- | 3496 |
| Pediatricians | --- | 2561 |
| Registered nurses | 1,027,135 | --- |
| Nursing assistive personnel[c] | 247,857 | --- |
| Nurses in contact with patients | --- | 51,634 |
| Bassinets | 37,871 | 2,460 |
| Hospitals with neonatal intensive care units (NICU) | NA | 55 |
| NICU beds | 17,388 | --- |
| Incubators | --- | 1,406 |
| Infant radiant warmers | --- | 415 |
| Delivery beds | --- | 268 |
| Hospitals with electronic health record | | |
| Not implemented | 12 | 42 |
| Partially implemented | 290 | --- |
| Fully implemented | 1,045 | --- |
| Implemented[d] | --- | 112 |
| Not reported | 54 | --- |

— Indicates variables were not available for that dataset.

[a]US dataset was restricted to hospitals that provides obstetric care, ≥500 annual births, ≥1 physician, ≥1 registered nurse, and ≥1 bassinet.

[b]Mexican dataset was restricted to Mexican Social Security Institute (IMSS) hospitals with ≥500 annual births, ≥1 obstetrician, ≥1 pediatrician, ≥10 nurses in contact with patients, and ≥1 bassinet.

[c]Included certified nursing assistant or equivalent unlicensed staff assigned to patient care units and reporting to nursing.

[d]Data on electronic medical record implementation in Mexico did not specify whether partially or fully implemented.

Table 4. In the US for all three obstetric levels, we ran two regression models, one that included hospitals with less than the median annual births from the sample and one that included hospitals with greater than the median annual births. The dependent variable of both models was hospital annual births and independent variables included number of nurse providers at the hospitals, number of bassinets in the hospital, and number of NICU beds in the hospital. For Mexico, to estimate nurse providers we had two regressions, one with a dataset that included hospitals with less than the median number of annual births for the sample and one that included hospitals with greater than the median annual births. Each regression model had annual births as the dependent variable and included the independent variables of nurse providers as the hospital, bassinets and infant radiant warmers.

**Action 2** determined how many of each type of healthcare professional was needed to be trained. This was estimated by dividing the healthcare professionals' coefficients from the regression analyses in Action 1 by the number of daily births. In hospitals that perform many cesarean-sections, births tend to fall on the weekdays [47]. This was confirmed by data from

**Table 3. Actions for estimating the annual training costs of BFHI at the hospital level in the United States and Mexico.**

| Action | Aim | Process | Data used | Model specification | Notes |
|---|---|---|---|---|---|
| 1 | We ran Ordinary Least Square regressions to estimate the coefficients of association between different types of healthcare providers and annual births in hospitals with obstetric care. | We identified the following variables in the data set/s: -annual births -total number of key healthcare providers that could/are involved in the maternity unit in the hospital (i.e. around patients when initiating breastfeeding or breastfeeding postpartum) -covariates that inflated the number of providers that need to be trained (i.e. number of NICU beds) -covariates associated with annual births (i.e. number of basinets) We created separate regressions for each type of provider, in which the dependent variable was annual births. | **United States:** American Hospital Association, 2014 [36] **Mexico:** Government of Mexico Health Resources, 2018 [37] | **United States:** $Annual\ Births_i = \alpha_i + \beta_{1i}(HC\ provider) + \beta_{2i}(NICU\ beds) + \beta_{3i}(Bassinets) + \varepsilon$ where the annual births in hospital $i$ is a function of $\beta_1$ number of healthcare providers in hospital $i$, $\beta_2$ is number of NICU beds in hospital $i$, and $\beta_3$ is number of bassinets in hospital $i$. Healthcare providers were defined as: (i) primary care physicians (general practitioner, general internal medicine, family practice, general pediatrics, obstetrics, geriatrics), (ii) hospitalists, (iii) registered nurses, (iv) nursing assistive personnel. **Mexico:** $Annual\ Births_i = \alpha_i + \beta_{1i}(HC\ provider) + \beta_{2i}(Infwarmers) + \beta_{3i}(Bassinets) + \varepsilon$ where the annual births in hospital $i$ is a function of $\beta_1$ number of healthcare providers in hospital $i$, $\beta_2$ is number of infant radiant warmers in hospital $i$, and $\beta_3$ is number of bassinets in hospital $i$. Healthcare providers were defined as: (i) gynecologists, (ii) pediatricians, (iii) nurses. | Independent variables will vary according to the country health system context and data availability. |
| 1A | We tailored Action 1 to relevant hospital characteristics (i.e., annual births, specialty level) to improve model predictions. | We found adequate ways of describing the level of specialization in the obstetric care given and size/intensity of maternity unit such as: -categorical variables identifying the type of specialty level -births below/above the median annual number of hospital births We created separate regressions using such categorizations of hospitals (which can be combined, i.e. highly specialized obstetric care in hospitals with annual births below and above the median of annual births). | **United States:** American Hospital Association, 2014 [36] **Mexico:** Government of Mexico Health Resources, 2018 [37] | **United States:** $Annual\ Births_{ijz} = \alpha_i + \beta_{1ijz}(HC\ provider) + \beta_{2ijz}(NICU\ beds) + \beta_{3ijz}(Bassinets) + \varepsilon$ where $j$ is the specialty level of the hospital and $z$ is the size of annual births. Specialty level was defined by a preestablished variable in the dataset with three levels: (i) level 1, provided services for uncomplicated maternity and newborn cases, (ii) level 2, provided service for all uncomplicated and most complicated cases, and (ii) level 3, provided services for all serious illnesses and abnormalities. For each level, the regressions also accounted for size based on the median of annual births: We categorized hospitals as larger or smaller than the median number of hospitals. For level 1, 2 and 3 respectively the rounded medians were: 800, 1300, and 2800. The combination of these to variables led to estimating 6 specific regressions for each type of provider. **Mexico:** $Annual\ Births_{iz} = \alpha_i + \beta_{1iz}(HC\ provider) + \beta_{2iz}(NICUInfwarmers) + \beta_{3iz}(Bassinets) + \varepsilon$ where $z$ is the size in terms of annual births. Size of the hospital was based on the median of annual births: We categorized hospitals as larger or smaller than the median, which was 2000. The combination of these to variables led to estimating 2 specific regressions for each type of provider. | Characterization of the hospitals will depend on the health system and available data. |
| 2 | We determined the number of each type of provider that needed to be trained per hospital and assumed that births occurred more frequently on weekdays (were non-uniform in frequency). | We estimated the health providers needed per day (estimating it yearly would inflate the number) -by dividing the annual births by 255 days, assuming that most births happened on weekdays, to get the daily average number of births (daily births) -then we divided daily births by $\beta_1$ -rounded the obtained numbers to the next integer (not doing so would underestimate the required resources) | **United States:** American Hospital Association, 2014 [36] **Mexico:** Government of Mexico Health Resources, 2018 [37] | $providers\ trained_i = \frac{annual\ births/261}{\beta_1}$ Do this for each provider type from each scenario-specific regression, $\beta_{1ijz}$ in the case of the United States, and $\beta_{1iz}$ for Mexico. | These estimations assumed non-uniform distributions of births over the year, with the majority births occurring during weekdays and not on weekends (other assumptions could be modelled adapting to contextual information). |
| 3 | We calculated the replacement wages ($RW$) while health providers are being trained. | For each type of provider, we multiplied the number of health providers that needed to be trained from Action 2 (or Action 2A) by -provider's specific salary -number of total training hours | **United States:** American Hospital Association, 2014 [36]; Occupational Employment Statistics (OES) 2019 [38]; First Latch Training Costs, 2020 [43] **Mexico:** Government of Mexico Health Resources, 2018 [37]; Mexican Institute of Social Security-National Union of Social Security Workers, 2019 [39] | $RW_i = [(providers\ trained_i) \times (hourly\ wage) \times (hrs\ BFHI\ training)]$ where hourly wages are specified as the average wage per type of provider and hours of training may be different by type of providers. *Training assumptions:* **United States** (online and clinical) -primary care physicians: 9 hours of training, 1 hour of clinical training -registered nurses: 15 hours of training, 5 hours of clinical training -nursing assistive personnel: 3 hours of training. **Mexico** (face-to-face and clinical) - all providers: 14 hours of training, 6 hours of clinical training | If there are no local BFHI training courses available, UNICEF BFHI generic training can be used to estimate hours. Regional estimations could also be performed if wage data is available. |

*(Continued)*

**Table 3.** (Continued)

| Action | Aim | Process | Data used | Model specification | Notes |
|---|---|---|---|---|---|
| 4 | We estimated the direct training module costs ($TM$) per provider. | For each type of provider, we multiplied the number of health providers that need to be trained from Action 2 (or Action 2A) by -the cost of the training module (adjust per type of provider if needed) | **United States:** American Hospital Association, 2014 [36]; First Latch Training Costs, 2020 [43] **Mexico:** Government of Mexico Health Resources, 2018 [37]; Asociación de Consultores de Lactancia Materna (ACCLAM)[44] | $TM_i = [(providers\ trained_i) \times (C\ BFHI\ training)]$ where ($C\ BFHI\ training$) is the cost of training | If training costs and wages are not from the same year, make sure to adjust them to reflect same year monetary value. |
| 5 | We calculated the education monitoring system cost ($MS_i$) per hospital. | For each hospital we searched for proxy variables of hospital technology (i.e., electronic medical records). Based on the proxy technology variable we categorized hospitals with or without technology -hospitals with technology will only require installing training files into the pre-existing monitoring system, which will have a one-time fee -hospitals without technology will require initial and fixed costs. | **United States:** American Hospital Association, 2014 [36]; MS price quotes[a] **Mexico:** Government of Mexico Health Resources, 2018 [37]; MS price quotes[a] | $MS_i = (Initial\ costs) + (Fixed\ costs)$ where ($Initial\ costs$) = 0 and ($Fixed\ costs$) = 0 if the hospital already has a technology system in place and would only have a file installation one-time fee of USD 75 per hour. For United States and Mexico, we assumed 3 hours of installation. For United States and Mexico, the same source was used to estimate ($MS_i$) ($Initial\ costs$) = 5,000 ($Fixed\ costs$) = 2,000 | If available, identify ($MS_i$) in local market prices. |
| 6 | We computed ($TC$) by adding ($RW$), ($TM$), and ($MS$), (Actions 3–5) for a final BFHI training cost estimate. | We performed a summation of all the components of ($TC$), for all hospitals within a defined category (i.e., depending on level of specialization and size) | Information from Actions 3–5 | $CT_t = \sum_i^n [RW_i + TM_i + MS_i]$ where $CT_i$ is the total cost of training hospitals of type $t$, and is the summation of ($RW$), ($TM$), and ($MS$), for hospital $i$ to $n$ | |

[a]Price quotes for the education monitoring systems were obtained in July 2020 from a free US software advisory service and included pricing from US companies like (but not limited to) Bridge and Skyprep.

Mexico [48] (data presented in S1 Fig). Therefore, computations were based on weekdays (261 days) instead of full weeks (365 days).

In addition, we modelled two real world scenarios (data presented in S1 Table) so that stakeholders could assess which condition most closely resembled their own. The first scenario assumed that births occurred on all days of the week including weekends: Computations were based on 365 days rather than 261 days as we assumed in Table 3. This provided a lower bound estimate of BFHI training cost per birth because it assumed fewer births per day and correspondingly fewer staff. Our second scenario doubled the number of providers to account for high end estimates of training costs. Since some months of the year have higher birth volume (data presented in S2 Fig), more staff would be needed to accommodate the higher number of patients. The final estimates from these scenarios are presented in S1 Table.

**Action 3** calculated the replacement wages (RW), which is the multiplication of the number of staff that needed training (calculated in Action 2) by staff salaries and by the number of hours required for training. We used additional data sources to obtain these values: For US salary estimates, we used the Occupational Employment Statistics database, which utilized national data from May 2019 [38]. For Mexico's salary estimates, we used the 2019 IMSS labor union contractual agreement, where salaries are published [39]. Salary estimates from both countries were adjusted to the cost year 2020 using the Consumer Price Index for all goods

**Table 4. Estimates of the number of healthcare providers involved in maternal care by obstetric level and hospital size in the two countries included in the BFHI training costs.**

| United States | Hospital level 1[a] | | Hospital level 2[a] | | Hospital level 3[a] | |
|---|---|---|---|---|---|---|
| | <800 annual births (n = 144) | ≥800 annual births (n = 168) | <1300 annual births (n = 297) | ≥1300 annual births (n = 340) | <2800 annual births (n = 221) | ≥2800 annual births (n = 231) |
| Registered nurses | 9635 | 2136 | 2883 | 4642 | 12,503 | 13,820 |
| Assistive nurses[b] | 2146 | 1420 | 1789 | 4988 | 2872 | 6280 |
| Physicians | 1749 | 502 | 1626 | 1954 | 1091 | 1844 |
| Hospitalist | 382 | 964 | 448 | 597 | 899 | 667 |
| **Mexico** | **Hospitals ≤2,000 annual births** (n = 73) | | **Hospitals >2,000 annual births** (n = 81) | | | |
| Nurses | 123 | | 440 | | | |
| Gynecologist/ Obstetricians | 73 | | 84 | | | |
| Pediatricians | 73 | | 84 | | | |

[a]Level 1 of obstetric care, provided services for uncomplicated maternity and newborn cases; level 2 provided service for all uncomplicated and most complicated cases; and level 3, provided services for all serious illnesses and abnormalities.

[b]Included certified nursing assistant or equivalent unlicensed staff assigned to patient care units and reporting to nursing.

and services [49]. To determine the hours of training needed in the US, we used the Baby Friendly USA Guidelines [46]. These guidelines required primary care physicians receive at minimum 3 hours of mandatory BFHI training and registered nurses receive 15 hours of training and 5 hours of clinical training. For nursing assistive personnel (defined using the AHA definition, as certified nursing assistants or equivalent unlicensed staff assigned to patient care units and reporting to nursing), training hours were the same as performed for a popular US, outsourced training course, First Latch, which we describe in further detail below. For Mexico, training hours were derived from the National Association of Certified Lactation Consultants (ACCLAM for its acronym in Spanish [44]), which offered a face-to-face course of 14 theoretical, face-to-face hours and 6 clinical hours for all providers (pricing information was received by the authors via email).

**Action 4** estimated the direct training costs (TM), for which we multiplied the average number of staff to be trained by the cost of the training per staff member. For this calculation we used prices of local, BFHI pre-designed training modules. As mentioned above we used First Latch [50] for US data, which has trained approximately 70% of US hospitals (per discussion with a First Latch salesperson) who were currently BFHI-designated at the time of the study. For Mexico, we used prices from ACCLAM. We chose to use local training courses since they are designed specifically for the countries' unique healthcare systems and because the courses consider economies of scale by providing lower prices for training larger numbers of staff.

**Action 5** added the cost of an education management system—a computer software platform that stores online BFHI training modules and tracks staff training. Hospitals that already had education monitoring systems, only required a small fee (~USD 75) to install BFHI pre-designed training modules or set up course tracking. If hospitals had no monitoring system, we added an initial price of USD 5000 with a recurring USD 2000 annual fee, which came to USD 7000 for the first year. These prices were obtained by getting quotes from popular US companies (e.g., Bridge and SkyPrep) that sell education monitoring systems around the world. There were no data, however, for either country, on whether hospitals had education management systems already implemented. Therefore, we used a proxy variable of hospital technological capacity, which was whether hospitals had implemented electronic medical

records (e-records). If the hospital had fully implemented e-records, we assumed they also had an education management system, but if the hospital had no e-records or partially implemented e-records, we assumed they did not. Under our assumption, hospitals with e-records would have a cost of USD 75 plus USD 2000, while those without, would have a cost of USD 7000, which is the price of purchasing a new education management system. Since our analysis was only for the initial training, the recurring cost was only calculated for the first year.

**Action 6** is the final step, which computed TC by summing RW, TM and MS (Actions 3–5) for a final BFHI training cost estimate. For the purpose of this analysis, we calculated the full cost as well as the per birth cost of staff training by dividing the final cost estimate by the total annual number of births per hospital. We presented all wages and final costs in purchasing power parity (PPP). This method allows assessments between standards of living of different countries based on a basket of goods approach and allows for comparison of US and Mexico currencies. To calculate the costs in PPP, we used the purchasing power parities rate for Mexico 2019 from the OECD National Accounts Statistics database [51]. The PPP rate is 9.65 in national currency (Mexican peso) per USD, so wages and costs in pesos were divided by the PPP rate.

All calculations and statistical analyses were performed in Stata, version 15 (StataCorp, College Station, USA).

## Results

The US AHA dataset [36] had n = 6240 total hospitals and healthcare facilities; n = 2430 (39%) of those indicated they had obstetric care; and n = 1591 (66%) of those had over 500 births. The Government of Mexico Health Resources dataset [37] had n = 2922 urban and rural public clinics, and primary, secondary and tertiary level public hospitals; n = 395 (14%) hospitals belonged to the social security system and n = 154 (39%) of those were affiliated with IMSS and provided obstetric care (with over 500 births).

Table 4 provided the estimates of staff to be trained for each type of healthcare provider by hospital size and level of obstetric care. The US had more providers that needed training compared to Mexico, with nurses as most common provider.

Using the staff estimates from Table 4, we calculated the total cost of implementing Step 2 of the BFHI in Tables 5–8. Table 5 presented the salaries and pricing we obtained from outsourced vendors for each costing factor (replacement wages, training costs, and monitoring system costs). Tables 6 and 7 applied those costs to hospital size and level of obstetric care for the US and Mexico. Table 6 presented a minimal and comprehensive level of training costs for the US. Table 7 presented the costs for Mexico.

Using Tables 6 and 7, we calculated the proportional contribution of the three costing factors to the total BFHI training: On average in the US, 80% of the total costs were the replacement wages, compared to 19% in Mexico. The direct training costs were similar in the US and Mexico, 10–11% of the total costs. The education monitoring system was the most significant relative contribution to total costs in Mexico at 72%, compared to only 9% in the US.

Table 8 presented the total cost of implementing BFHI Step 2 per birth in each country. Costs per birth varied substantially between countries. In the US, the cost per birth ranged from USD 7.27 (which were obstetric level 2, large hospitals with minimal training) to USD 125.39 (which were obstetric level 1, small hospitals with comprehensive training). In Mexico, the costs ranged from PPP 2.68 (large hospitals) to PPP 6.14 (small hospitals). S1 Table presents the training cost per birth for two additional scenarios. The first scenario assumes that births were uniformly distributed across the week, not just occurring on weekdays as we assumed earlier. The second is assuming a random distribution of births throughout the year hence doubling the staff to provide a high-end cost estimate.

**Table 5. Costs (in USD) of the 3 factors used for estimating total BFHI training costs.**

| | United States | | | | Mexico | | |
|---|---|---|---|---|---|---|---|
| **Staff replacement wages per healthcare professional** | | | | | | | |
| | Wages per hour in 2019 (USD) | Minimum training (hours) | Staff replacement wages (USD) | | Wages per hour (USD[a], PPP[b]) | Training (hours) | Staff replace-ment wages (USD[a], PPP[b]) |
| Registered nurses | 37 | 20 | 740 | Nurses | 4, 9 | 20 | 82, 190 |
| Assistive nurses[c] | 17 | 3 | 51 | | | | |
| Physicians | 101 | 3 | 306 | Physicians | 8, 19 | 20 | 160, 374 |
| Hospitalist | 101 | 3 | 306 | | | | |
| **Training module prices per number of health professional enrolled[d]** | | | | | | | |
| Registered Nurses[e] | | | | All professionals (USD, PPP[b] per person)[f] | | | 62, 144 |
| 1 | 134.00 | | | | | | |
| 2–9 | 94.00 | | | | | | |
| 10–49 | 86.00 | | | | | | |
| 50–100 | 81.00 | | | | | | |
| 101–150 | 76.00 | | | | | | |
| 151–200 | 72.00 | | | | | | |
| ≥201 | 69.00 | | | | | | |
| Nursing Assistive personnel | | | | | | | |
| | Minimal Training[g] | Comprehensive Training[h] | | | | | |
| 1 | 25.00 | 110.00 | | | | | |
| 2–9 | 20.00 | 90.00 | | | | | |
| 10–49 | 15.00 | 75.00 | | | | | |
| ≥50 | 15.00 | 65.00 | | | | | |
| Physicians | | | | | | | |
| | Minimal Training[j] | Comprehensive Training[j] | | | | | |
| 1 | 51.00 | 136.00 | | | | | |
| 2–9 | 40.00 | 110.00 | | | | | |
| 10–49 | 40.00 | 100.00 | | | | | |
| 50–100 | 37.00 | 87.00 | | | | | |
| **Education monitoring system[k]** | | | | | | | |
| Initial[l] | 5000.00 | | | Initial[l] | 5000.00 | | |
| Ongoing[m] | 2000.00 | | | Ongoing[m] | 2000.00 | | |
| Installation fee[l] | 225.00 | | | Installation fee[n] | 225.00 | | |

[a] Exchange rate of Mexican peso to US dollar 22.5

[b] Organisation for Economic Co-operation and Development (OECD) purchasing power parity (PPP) indicator for 2019.

[c] Included certified nursing assistant or equivalent unlicensed staff assigned to patient care units and reporting to nursing.

[d] Pricing for training modules was received upon request via email for the United States [50] and Mexico [44].

[e] Registered nurses had 15 hours of online training and 5 hours of clinical training.

[f] All professionals received 14 hours of face-to-face training and 6 clinical hours.

[g] Minimal training for assistive nurses included 3 hours of online training. [h] Comprehensive training for assistive nurses included the minimal training (3 hours of online training), plus another 6 hour online module "Basics of lactation management" and 1 clinical hour.

[i] Minimal training for physicians included 3 hours of online training. [j] Comprehensive training for physicians included the minimal training (3 hours of online training), plus another 6 hour online module "Basics of lactation management" and 1 clinical hour.

[k] Price quotes for the education monitoring systems were obtained in July 2020 from a free US software advisory service and included pricing from US companies like (but not limited to) Bridge and Skyprep.

[l] This is a one-time cost only in year 1 for purchasing the system.

[m] This cost reoccurs yearly beginning with year 1 as a maintenance cost to the system.

[n] This is a one-time cost for hospitals that already have their own education monitoring systems. It is the price to install online modules and breastfeeding course tracking in the system already in place. This software installation fee was USD 75 per hour, and we assumed 3 hours.

**Table 6. First-year implementation cost estimates (in USD) for BFHI training in the United States (in USD) by obstetric level and hospital size.**

| United States | Hospital level 1[a] | | Hospital level 2[a] | | Hospital level 3[a] | |
|---|---|---|---|---|---|---|
| | <800 annual births (n = 144) | ≥800 annual births (n = 168) | <1300 annual births (n = 297) | ≥1300 annual births (n = 340) | <2800 annual births (n = 221) | ≥2800 annual births (n = 231) |
| *Minimal training scenario[b]* | | | | | | |
| Wage replacement[c] | 7,926,288 | 2,104,118 | 2,862,297 | 4,476,759 | 10,056,305 | 11,365,320 |
| Direct training[c] | 923,880 | 276,667 | 386,846 | 595,121 | 1,174,043 | 1,353,635 |
| Education monitoring system[d] | 375,400 | 310,800 | 661,825 | 699,500 | 357,725 | 401,975 |
| Total | 9,225,569 | 2,691,585 | 3,910,968 | 5,771,380 | 11,588,074 | 13,120,930 |
| *Comprehensive training scenario[b]* | | | | | | |
| Wage replacement[c] | 9,676,616 | 3,301,642 | 4,530,721 | 6,855,354 | 11,792,471 | 13,867,084 |
| Direct training[c] | 1,189,665 | 472,017 | 657,256 | 1,093,301 | 1,497,908 | 1,894,925 |
| Education monitoring system[d] | 375,400 | 310,800 | 661,825 | 699,500 | 357,725 | 401,975 |
| Total | 11,241,681 | 4,084,459 | 5,849,802 | 8,648,155 | 13,648,104 | 16,163,984 |

[a]Level 1 of obstetric care provided services for uncomplicated maternity and newborn cases; level 2 provided service for all uncomplicated and most complicated cases; and level 3 provided services for all serious illnesses and abnormalities.

[b]For physicians and nursing assistive personnel, minimal training was 3 hours of online training whereas comprehensive training was 3 hours of online training plus another 6-hour online module "Basics of lactation management" and 1 clinical hour; For registered nurses, minimal and comprehensive training was the same (see Table 5).

[c]Wage replacement and direct training costs were calculated using the comprehensive/maximum number of training hours from Table 5.

[d]Education monitoring system costs were calculated using the prices from Table 5 based on whether hospitals already had electronic monitory systems or not (assessed by the proxy variable of having electronic records or not).

## Discussion

This is the first study to take a hospital system-wide approach to estimate Step 2 costs in the US and Mexico. The cost of BFHI staff training was between USD 7.27–125.39 per birth in the US and PPP 2.68–6.14 per birth in Mexico, depending on hospital size, obstetric care level, number of staff, and minimum vs. maximum training hours (Table 8).

Dellifraine et al. [22] estimated first-year training costs as USD 15,493 (adjusted for inflation from 2013 to 2020 using the Consumer Price Index [49]) in a Southwestern hospital in the US with 2800 annual deliveries, equaling USD 5.5 per birth. Our method, which calculated

**Table 7. First-year implementation cost estimates (in USD) for BFHI training in Mexico by hospital size.**

| Mexico | Hospitals ≤2000 annual births (USD[a], PPP[b]) n = 73 | Hospitals >2000 annual births (USD[a], PPP[b]) n = 81 |
|---|---|---|
| Wage replacement | 33,429, 77,985 | 62,836, 146,588 |
| Direct training | 16,633, 38,803 | 37,595, 87,703 |
| Education monitoring system[c] | 128,425, 299,596 | 200,225, 467,095 |
| Total | 178,487, 416,384 | 300,656, 701,386 |

[a] Exchange rate of Mexican peso to US dollar 22.5

[b]Organisation for Economic Co-operation and Development (OECD) purchasing power parity (PPP) indicator for 2019.

[c]Education monitoring system costs were calculated using the prices from Table 5 based on whether hospitals already had electronic monitoring systems or not (assessed by the proxy variable of having electronic records or not).

**Table 8. Total hospital births and estimates of cost per birth (in USD) for BFHI training in the United States and Mexico.**

| United States | Hospital level 1[a] | | Hospital level 2[a] | | Hospital level 3[a] | |
|---|---|---|---|---|---|---|
| | <800 annual births (n = 144) | ≥800 annual births (n = 168) | <1300 annual births (n = 297) | ≥1300 annual births (n = 340) | <2800 annual births (n = 221) | ≥2800 annual births (n = 231) |
| Total annual births | 91,081 | 255,582 | 265,277 | 826,332 | 392,290 | 1,049,076 |
| Mean births per weekday (SD) | 2.42 (0.33) | 5.83 (3.86) | 3.42 (0.87) | 9.31 (4.74) | 6.80 (2.37) | 17.40 (6.85) |
| Mean cost per birth with minimal training (SD)[b] | 101.49 (5.65) | 11.01 (2.8) | 15.09 (4.29) | 7.18 (1.63) | 29.84 (2.23) | 12.59 (0.79) |
| Mean cost per birth with comprehensive training (SD)[b] | 123.65 (5.69) | 16.61 (2.94) | 22.43 (4.37) | 10.72 (1.67) | 35.19 (2.39) | 15.51 (0.81) |
| **Mexico** | **Hospitals ≤2000 annual births** (n = 73) | **Hospitals >2000 annual births** (n = 81) | | | | |
| Total annual births | 76,217 | 307,969 | | | | |
| Mean births per weekday (SD) | 4.00 (1.71) | 14.57 (8.63) | | | | |
| Mean cost per birth (SD) | 2.63 (3.60) | 1.15 (1.16) | | | | |
| Mean cost per birth, PPP[c] | 6.14 | 2.68 | | | | |

[a]Level 1 of obstetric care provided services for uncomplicated maternity and newborn cases; level 2 provided service for all uncomplicated and most complicated cases; and level 3 provided services for all serious illnesses and abnormalities.

[b]Wage replacement and direct training costs were calculated using the minimal and comprehensive number of training hours, respectively, from Table 5.

[c]Organisation for Economic Co-operation and Development (OECD) purchasing power parity (PPP) indicator for 2019 in USD.

the average training cost per birth in the US rather than for one hospital, estimated a slightly higher but comparable cost of USD 7.3 per birth (for hospitals with ≥1300 annual births, obstetric level 2 care and minimal training) (Table 8). Nersesyan [13], which was a micro-costing study in Jordan, estimated that training and monitoring cost USD 2.15 per woman served by the clinic (adjusted for inflation from 2005 to 2020 using the Consumer Price Index [49]), which was more similar to our estimate of USD 2.63 for staff training in Mexico in hospitals with less than 2000 annual births (Table 8).

Our method shared similarities and differed from the only other financial planning tool that exists to calculate BFHI costs, designed by the WBCi [21, 25]. This tool includes an interactive spreadsheet that sums a number of items as a budgeting aid for individual hospitals. Unlike our method, the WBCi included costs associated to training outside the health facility, so it included costs such as transportation, per diem and meals. Our method assumed that training occurred at the facility, so we did not include these costs. While the WBCi tool is a useful starting point, especially for hospital administrators to create budgets, our analysis went an extra step to estimate the average costs of implementing the BFHI at a larger scale, and it utilized publicly available data in a method that can be replicated, which is needed for policy-making at the national and subnational levels. One additional advantage is that our method includes replacement staff wages, which contributed to 80% of the total training costs in the US. These were omitted in the WBCi tool.

We highlight several additional takeaways from applying this method to the US and Mexico. In the US, hospitals with the lowest obstetric level of care (level 1) and with fewest annual births (<800 births per year) had the highest per birth training costs, USD 102–125 per birth (Table 8). It is possible that smaller hospitals have lower quality data, and in our dataset, obstetric level 1 hospitals had the smallest sample size. Future cost effectiveness analyses may show that small hospitals with low obstetric care specialization have lower cost effectiveness compared with larger hospitals that have specialized care perhaps because they do not have the advantage of economies of scale. Smaller, less specialized hospitals had fewer staff to train,

resulting in fewer course discounts. In addition, the low annual birth volume in these hospitals significantly inflated the cost per birth. One solution could be for small hospitals to split training costs with other small institutions and train together via videoconferencing, a strategy rural hospitals in Canada, the US and Australia have used successfully [52, 53].

Another key finding was that total training costs were significantly higher in the US compared to Mexico. This is because the US had more health professionals per 1000 people, 8% more physicians and 310% more nurses than Mexico per 1000 people. In addition, US provider salaries were 4–5 times more than Mexican providers (Table 5) [38, 39]. These differences impacted how much replacement wages affected the Step 2 implementation total cost: Across all hospitals in the US, replacement wages were 80% of the total BFHI training costs, compared to all hospitals in Mexico, where replacement wages were only 19% of the total costs. It is important to highlight that Mexico has medical staff per patient that are below the averages of the Organization for Economic Co-operation and Development (OECD), and this is believed to affect the quality of services [54]. Cost estimations in this study are based on the observed health system, rather than the optimal health system. Therefore, if health personnel to patient ratios are improved in the future, BFHI training costs will increase.

In addition, the prices of predesigned courses differed between countries. In Mexico, training courses from ACCLAM were PPP 144 for all health professionals, which were similar to the price for nurses in the US, which cost USD 134 per nurse. Training physicians in the US cost less than half of nurses (USD 51) because they had 3 plus 1 optional hour of training compared with 15 plus 5 clinical hours of training for nurses. For comparison, the UK UNICEF BFHI online training offerings, which are well-designed [31], were USD 27 per general practitioner—notably less than courses in the US and Mexico. This brief comparison between BFHI training vendors indicates that further research is needed to compile vendor costs and their content offerings to catalogue how Step 2 is being met across countries.

Furthermore, little is known about the efficacy of different course durations, content, and modes (clinical vs. practical skills) [14]. This is key to address because duration, for example, has a substantial impact on training costs. Baby Friendly USA only requires physicians to receive 3 hours of training, whereas in Mexico all healthcare professionals are expected to receive 20 hours of training, per WHO/UNICEF guideline recommendations. Further, to our knowledge, no studies have compared the effectiveness of utilizing the free WHO/UNICEF materials that allow hospitals to train staff in-house versus paying for an outsourced training course. A 2017 systematic review of Step 2 found staff knowledge increased after training, yet it only included six studies that sufficiently measured pre- and post-training staff knowledge [32]. Of those studies included, only one study from 1999 used the WHO/UNICEF course [55]. The review did not investigate how the type of training may have affected the outcomes, which leaves a critical gap in the literature.

Studies are also needed to assess the most efficient approach to training, whether online, in person or a hybrid of both. All BFHI WHO/UNICEF materials, which included PowerPoint slides, exercises, forms and checklists, require downloading or printing and thus are not designed to be used as an online module [56]. If hospitals were able to utilize hybrid or online BFHI training, it could decrease costs by reducing the amount of time needed for a trainer and the time needed to train trainers, the latter of which can be substantial–for example, it costs USD 10,200 to train twelve trainers using the UK BFHI training package. Staff replacement wages could be reduced if healthcare professionals could complete training on their mobile devices, for example, during down time in their shifts—a strategy that has been utilized increasingly in hospitals over the past two decades [57], including in remote sites [58] and in low- and middle-income countries [59]. These savings also apply for online refresher courses that recur annually, allowing for sizable long-term savings. The efficacy of online courses has

been shown to be equal [60] if not greater [61] than live instruction for improving health professional knowledge in a range of countries, including in rural areas in high income countries [62]. Specifically for breastfeeding training, a study in Mexico showed that a hybrid online/face-to-face model may be best to increase staff knowledge [63]—though further research is needed in other regions. Given that global internet penetration continues to increase rapidly, we recommend that hospitals, if feasible, move to a mix of online and in-person modalities. Research is needed across different countries and contexts to identify which aspects of training are best conducted online versus face-to-face. Future economic analyses are also needed to test the comparative cost-effectiveness of face-to-face, online, and hybrid breastfeeding courses [64].

This study had some limitations. First, it is possible that we underestimated the education monitoring system costs. Because there were no publicly available data on whether hospitals already had those systems in place, we assumed that if a hospital had electronic medical records that they also already had an education monitoring system. Second, we assumed the same price for the education monitoring system for the US and Mexico, even though there may be regional differences in installing the system. Third, we did not account for rural versus urban training cost differences. This, however, may be reflected in the number of staff, which we do account for. Fourth, we did not account for human resource constraints on scaling up for small hospitals. Another analysis for smaller hospitals that includes those with fewer than 500 births, would likely generate more accurate cost estimates for this specific group of small hospitals. Fifth, we assumed all training was performed in the hospital facility. Training outside of the facility can incur travel and per diem costs [32]—these, however, were very small (0.4% of the total BFHI training cost) in one US study [22], yet could vary depending on the location. Sixth, staff turnover can affect implementation costs [15] since newly hired staff require more costly training, which we did not take into consideration. Finally, since we estimated first year costs only, our method may limit the ability of stakeholders to advocate for funds beyond the first year. Training costs are likely lower after the first year since we assumed that no staff had received prior BFHI training, and in subsequent years, previously trained staff would only require refresher trainings, which are shorter. In addition, for hospitals with no electronic monitoring system, there would not be the initial investment costs.

Overall, our method offered a unique, macro-costing perspective of Step 2. Notable strengths include generalizability across many hospitals in a healthcare system and few model-inputs that make estimations simple for the user. No previous study to our knowledge has provided a way for policy-makers and stakeholders to estimate the cost of implementing the BFHI Step 2 at a country level, which this method allows for. Prior micro-costing approaches may have given more precise estimates for individual hospitals [13, 23], yet these studies provided no method for estimating Step 2 costs for a country or a system of hospitals, as ours does.

Given that Step 2 is the backbone of successful BFHI implementation, it is critical that policy-makers, hospital administrators and healthcare stakeholders are empowered to estimate staff training costs to obtain funding and to identify feasible and sustainable training models. Here we proposed a novel methodology to estimate the costs for a system of hospitals in two countries, the US and Mexico. Our method utilized publicly available data that hospital administrators and governments commonly collect, and the method required relatively few variables. If similar data exist in other countries, the method can be replicated once we validate it in the US and Mexico with real world data, which is our next step along with adding training cost estimations that extend past the first year. Overall, the novel approach presented here aimed to provide US and Mexican healthcare stakeholders with cost estimates of the BFHI training, one of the key cost-drivers of the entire BFHI program, to encourage successful and sustainable implementation.

## Supporting information

**S1 Fig. Births per day of the week from in 2018 and 2019 from birth certificates issued from the Mexican civil registry offices [43].**
(DOCX)

**S2 Fig. Births per day in 2018 and 2019 in Mexico from birth certificates issued from the Mexican civil registry offices [43].**
(DOCX)

**S1 Table. Sensitivity analyses assuming uniform weekly births and double the staff for total hospital births and cost per birth (in USD) for BFHI training in the United States and Mexico.**
(DOCX)

## Author Contributions

**Conceptualization:** Kendall J. Arslanian, Mireya Vilar-Compte, Graciela Teruel, Elizabeth C. Rhodes, Amber Hromi-Fiedler, Rafael Pérez-Escamilla.

**Data curation:** Kendall J. Arslanian, Mireya Vilar-Compte, Graciela Teruel, Annel Lozano-Marrufo, Rafael Pérez-Escamilla.

**Formal analysis:** Kendall J. Arslanian, Mireya Vilar-Compte, Graciela Teruel, Annel Lozano-Marrufo, Erika García.

**Funding acquisition:** Rafael Pérez-Escamilla.

**Investigation:** Kendall J. Arslanian, Mireya Vilar-Compte, Rafael Pérez-Escamilla.

**Methodology:** Kendall J. Arslanian, Mireya Vilar-Compte, Annel Lozano-Marrufo, Elizabeth C. Rhodes, Rafael Pérez-Escamilla.

**Project administration:** Kendall J. Arslanian, Mireya Vilar-Compte, Rafael Pérez-Escamilla.

**Resources:** Kendall J. Arslanian, Mireya Vilar-Compte, Rafael Pérez-Escamilla.

**Software:** Kendall J. Arslanian, Mireya Vilar-Compte, Graciela Teruel, Annel Lozano-Marrufo, Erika García, Rafael Pérez-Escamilla.

**Supervision:** Kendall J. Arslanian, Mireya Vilar-Compte, Graciela Teruel, Amber Hromi-Fiedler, Rafael Pérez-Escamilla.

**Validation:** Kendall J. Arslanian, Mireya Vilar-Compte, Graciela Teruel, Erika García.

**Visualization:** Kendall J. Arslanian, Annel Lozano-Marrufo.

**Writing – original draft:** Kendall J. Arslanian, Mireya Vilar-Compte, Graciela Teruel, Elizabeth C. Rhodes, Amber Hromi-Fiedler, Rafael Pérez-Escamilla.

**Writing – review & editing:** Kendall J. Arslanian, Mireya Vilar-Compte, Graciela Teruel, Annel Lozano-Marrufo, Elizabeth C. Rhodes, Amber Hromi-Fiedler, Erika García, Rafael Pérez-Escamilla.

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
