## [Decision Letter · Decision Letter 0]

7 Feb 2022

PONE-D-21-30699How much does it cost to implement the Baby-Friendly Hospital Initiative training step in the United States and Mexico?PLOS ONE

Dear Dr. Arslanian,

Thank you for submitting your manuscript to PLOS ONE. After careful consideration, we feel that it has merit but does not fully meet PLOS ONE’s publication criteria as it currently stands. Therefore, we invite you to submit a revised version of the manuscript that addresses the points raised during the review process.

The reviewers have made helpful suggestions as to how to strengthen the draft (below): please address these in the revised draft. 

We look forward to receiving your revised manuscript.

Kind regards,

Susan Horton

Academic Editor

PLOS ONE

Journal Requirements:

Reviewers' comments:

Reviewer's Responses to Questions

**Comments to the Author**

1. Is the manuscript technically sound, and do the data support the conclusions?

Reviewer #1: Partly

Reviewer #2: Yes

2. Has the statistical analysis been performed appropriately and rigorously? 

Reviewer #1: N/A

Reviewer #2: No

3. Have the authors made all data underlying the findings in their manuscript fully available?

Reviewer #1: Yes

Reviewer #2: Yes

4. Is the manuscript presented in an intelligible fashion and written in standard English?

Reviewer #1: Yes

Reviewer #2: Yes

5. Review Comments to the Author

Reviewer #1: This is a really interesting article that explores an economic modeling approach for step 2 of the BFHI program. The authors had a very clear grasp of the existing literature and the need for this type of research and modeling. The authors also clearly understood the tensions that exist between public health interventions and implementation through scale up processes. I believe with some additional clarification this article will be one that is cited for years to come.

Strengths:

The introduction was very through and provided a strong foundation for the purpose and need for the study. The authors were very clear in the formulas and regression approaches that were undertaken in their methodological approach. I really appreciated the table of steps as it was very easy to read and to follow. The data sources that were used were very appropriate and were adequately described. The economic approach was sound and the assumptions were reasonable for the model. The process is repeatable and the findings are in line with existing evidence. The study expands our current capability to cost BFHI step 2 for policymakers.

Weaknesses and Improvement Suggestions:

Greater clarity is needed in several areas. They are:

The methods section needs additional detail. It was unclear to me what cost year and method of inflation was used (e.g. consumer price index, medical cost index, general). It seemed like 2020 costs were used for US data while 2019 was used for Mexico for some costs and outcomes. The article would be improved if everything were standardized to the same year to the extent possible and if there was a appendix or supplemental table that more clearly described the year and standardization that occurred. The information is there I think, it is just not very clear.

The values that were used for staff salary and the year that salary represents is not clear. The authors do state they used wage data from the US bureau of labor statistics for the occupations. However, I am not sure what year/quarter this data was from, what the value was, and whether it was national or region specific. Table 5 shows cost data but I'm not sure if this is total or per hour. It would be helpful to add a supplemental table that more explicitly defines the estimates. I would encourage the authors to present the information as hourly wage and total. This is especially true for the US model because 80% of the cost was driven by labor replacement. I would urge the authors to also consider using an wage premium of 1.25 in there sensitivity analyses of wage to account for the likelihood that agency personnel would be needed for staff replacement. This would yield a stronger estimate as well as a better understanding of the sensitivity to wage fluctuation.

The sensitivity analyses are not well defined. The article would be stronger if the authors clarified what sensitivity modeling approach was used and how they varied cost/pricing. It would also help to know at what price points did the cost per birth and total change significantly (say 50% or 100% increase).

Finally, as a minor weakness, the article does need to be proof read an additional time. There were several places in several sections where phrases or words were duplicated (e.g. the the). One additional read through for those minor errors would make the study clearer and stronger.

Final Recommendation

This article is timely and explores a topic that is needed to improve our understanding and economic support for BFHI. I recommend that the authors review the article one more time to address the weaknesses identified above. I think once addressed, the study will be a significantly cited paper and provide a valuable resource to the BFHI community. I support publication with major revision.

Reviewer #2: This study is very important as training is a crucial part in the success of interventions. Given that BFHI has a crucial role in improving breastfeeding rate, providing the cost is essential for scale up, evaluation and/or advocacy purposes. I believe the costing method is pretty straight forward except for step 1 and 2 of the formula that I think might confuse readers. I have some clarification and comments that hopefully may improve the paper.

Methods:

1. There is a type o in the definition of variables of formula (1), page 7 line 141, averarage should be average.

2. Regarding the RW definition, wouldn’t it be better if it is defined the productivity loss associated with the training. For me the term replacement wage is a bit misleading, as whether the training takes place or not, the salary related to the service provided by the staff on training will be paid (to the staff or his/her replacement). The cost, however, stems from the work that is not done by the staff on training although she/he still receive their regular salary. In my opinion, the RW formula actually captures this productivity loss cost.

3. Do the staff on training receive some kind of extra benefits/incentive/per diem for joining the training? This really depends on the local regulation but may result in a quite a sum and should be calculated if this exists in the respective countries. I see that this study excluded this cost as it assumes that the training takes place in the hospital. But, again, this really depends on the local regulations and may need to be stated in the limitation or discussion section.

4. Page 8 line 171-172: the authors stated that the number of hospital staff since 2014 would not likely be substantially different. I am not familiar with the US system, so I would ask how can the authors be sure of such condition? It would be helpful if authors could provide 1 or 2 references to support this. In addition, the authors could also project the current number using some kind of available growth parameters for relevant variables?

5. I found it a bit difficult in following step 1 and 2 where there were regressions performed. I may have missed this, but how do exactly the coefficients obtained from the regressions were used in the overall calculation? Please elaborate.

Results

1. Page 19 line 303 – 304: I believe Step 2 here refers to the training, correct? Just for precaution, this may be misleading as readers may be confused between step 2 of the formula or step 2 as the training itself, even though there is already an explanation of step 2 of the BFHI (readers may have missed this).

2. I may have missed this, but were the costs between US and Mexico compared using PPP?

Discussion

1. Page 24 Line 392: I am not sure it is the place of the study to state that BHFI training had poor cost-effectiveness in small hospitals with low obstetric care specialization. Please consider whether the discussion should be on lower cost efficiency instead, which relates with the following arguments in the paragraph.

2. Page 25 line 406 – 407: this is important, please add references to support the statement.

3. Page 26 line 431. I agree that the future studies on cost effectiveness should be conducted, but I’m not sure whether the paragraph should stary in that context. The paragraph discusses on the more efficient approach of the training, which directly relates with the study. The need for cost effectiveness studies would be better placed at the end of the paragraph.

4. Page 27, limitations: I think the authors skipped the third limitation?

5. Limitations should also consider the possibility of differences in local characteristics which cannot be fully captured by the current method. This is actually the strength and weakness of micro costing which the authors mentioned in the background. However, it is also the weakness of the current method which can be used more generally, but may have missed local context variations.

6. PLOS authors have the option to publish the peer review history of their article (what does this mean?). If published, this will include your full peer review and any attached files.

Reviewer #1: No

Reviewer #2: No

---

## [Author Response · Author response to Decision Letter 0]

1 Jul 2022

Manuscript PONE-D-21-30699

Response to Reviewers 

Dear PLOS One Editorial Board,

Thank you to the editor for the opportunity to revise this manuscript. We were pleased to see the reviewers’ enthusiasm for the paper, and we grateful we were for their thorough and thoughtful comments. Below we have a point by point response (in red) to the reviewer comments (in black).

Reviewer #1 (Comments to the Author):

This is a really interesting article that explores an economic modeling approach for step 2 of the BFHI program. The authors had a very clear grasp of the existing literature and the need for this type of research and modeling. The authors also clearly understood the tensions that exist between public health interventions and implementation through scale up processes. I believe with some additional clarification this article will be one that is cited for years to come.

Strengths:

The introduction was very through and provided a strong foundation for the purpose and need for the study. The authors were very clear in the formulas and regression approaches that were undertaken in their methodological approach. I really appreciated the table of steps as it was very easy to read and to follow. The data sources that were used were very appropriate and were adequately described. The economic approach was sound and the assumptions were reasonable for the model. The process is repeatable and the findings are in line with existing evidence. The study expands our current capability to cost BFHI step 2 for policymakers.

Weaknesses and Improvement Suggestions:

Greater clarity is needed in several areas. They are:

The methods section needs additional detail. It was unclear to me what cost year and method of inflation was used (e.g. consumer price index, medical cost index, general). It seemed like 2020 costs were used for US data while 2019 was used for Mexico for some costs and outcomes. The article would be improved if everything were standardized to the same year to the extent possible and if there was a appendix or supplemental table that more clearly described the year and standardization that occurred. The information is there I think, it is just not very clear.

We added a sentence on line 262, page 17 to clarify the cost year and method of inflation. We also added the method and cost year to all places where we adjusted for inflation of training costs from other studies (page 24 line 392 and 398). Finally we updated costs to the same year of 2019 in Table 1, where possible, to address the issue of having different cost outcomes for the countries.

The values that were used for staff salary and the year that salary represents is not clear. The authors do state they used wage data from the US bureau of labor statistics for the occupations. However, I am not sure what year/quarter this data was from, what the value was, and whether it was national or region specific. Table 5 shows cost data but I'm not sure if this is total or per hour. It would be helpful to add a supplemental table that more explicitly defines the estimates. I would encourage the authors to present the information as hourly wage and total. This is especially true for the US model because 80% of the cost was driven by labor replacement. I would urge the authors to also consider using an wage premium of 1.25 in there sensitivity analyses of wage to account for the likelihood that agency personnel would be needed for staff replacement. This would yield a stronger estimate as well as a better understanding of the sensitivity to wage fluctuation.

Thank you for this comment. We added a sentence specifying the region, month and year from which we reported United States’ wages on page 17, line 260. 

We also incorporated the reviewer’s astute comment about presenting additional information in Table 5: We added provider hourly wages and hours of training to show how we calculated staff wage replacement in Table 5. 

The reviewer also suggested we account for staff replacement by conducting a sensitivity analysis with a wage premium of 1.25. However, the model already accounted for the wages of replacement staff who are needed to perform the regular duties of staff who were being trained (line 133, page 6; lines 145-149, page 7).

As the reviewer suggested, a sensitivity analysis may yield results that are more accurate due to wage fluctuations. We took a different approach to account for fluctuating staff needs through assessing two ‘scenarios’: the first scenario was increasing staff due to increasing birth volume on the weekdays, and the second was doubling the staff needed (to ensure that months with high birth volumes were sufficiently staffed, see S2 Figure below). 

The sensitivity analyses are not well defined. The article would be stronger if the authors clarified what sensitivity modeling approach was used and how they varied cost/pricing. It would also help to know at what price points did the cost per birth and total change significantly (say 50% or 100% increase).

We thank the reviewer for this comment. On page 16, line 247 we changed the language from performing “sensitivity analyses” to the “modelling scenarios”, which better represents our analytic intentions.

As such, we did not vary cost/price to generate price points where we can estimate percent increases in the cost per birth. Rather, we altered assumptions about birth volume during the week that replicate real world scenarios, which ultimately was the purpose of this paper. In our first scenario (line 248, page 16), we assumed that women gave birth in hospitals at an even rate throughout the week. This was in contrast to our initial assumption in Table 3 that births occurred primarily on weekdays [1]. In our second scenario (line 252, page 16), we doubled the number of staff who needed training, to ensure that there were sufficient staff in months where birth volumes were higher (see S2 Figure below).

We added a more thorough explain on page 16, lines 247-255 so that readers understand the intended purpose of including these estimates and to clarify that we altered assumptions to generate cost estimates for two scenarios rather than performed sensitivity analyses to identify price point changes.

S2 Figure. Births per day in 2018 and 2019 in Mexico from birth certificates issued from the Mexican Civil Registry Offices [2].

an=2,162,432 (n=125 were removed due to missing values or data entry errors). bn=2,092,142 (n=96 were removed due to missing values or data entry errors).

Finally, as a minor weakness, the article does need to be proof read an additional time. There were several places in several sections where phrases or words were duplicated (e.g. the the). One additional read through for those minor errors would make the study clearer and stronger.

Thank you for this comment. We did a close edit of the manuscript.

Reviewer #2 (Comments to the Author):

1. There is a type o in the definition of variables of formula (1), page 7 line 141, averarage should be average.

Thank you. We corrected this typo.

2. Regarding the RW definition, wouldn’t it be better if it is defined the productivity loss associated with the training. For me the term replacement wage is a bit misleading, as whether the training takes place or not, the salary related to the service provided by the staff on training will be paid (to the staff or his/her replacement). The cost, however, stems from the work that is not done by the staff on training although she/he still receive their regular salary. In my opinion, the RW formula actually captures this productivity loss cost.

Thank you for this comment. From an economic perspective, productivity loss occurs when one is not working on duties required of them. While providers cannot perform their required jobs during training (incurring a short term loss of productivity), in the long term, training in breastfeeding supportive skills produces better patient outcomes [3], outweighing the short term productivity loss during training. As readers may have similar thoughts on this point to the reviewer, we added this explanation on page 7, line 146.

3. Do the staff on training receive some kind of extra benefits/incentive/per diem for joining the training? This really depends on the local regulation but may result in a quite a sum and should be calculated if this exists in the respective countries. I see that this study excluded this cost as it assumes that the training takes place in the hospital. But, again, this really depends on the local regulations and may need to be stated in the limitation or discussion section.

When training is conducted outside of the facility, staff may receive per diem and travel funds for training. Dellifraine [4] estimated per diem and travel as a one-time USD 1500 start-up cost, which was 0.4% of the total training cost. Therefore, to make our method as simple as possible for the user, we decided to omit this cost and focus on training in the facility. We added this a limitation on line 487, page 28. In future refinements of this method, we may target countries that train primarily outside of facilities as case studies, but even so, we expect that the per diem and travel may be a small proportion of the total costs. 

4. Page 8 line 171-172: the authors stated that the number of hospital staff since 2014 would not likely be substantially different. I am not familiar with the US system, so I would ask how can the authors be sure of such condition? It would be helpful if authors could provide 1 or 2 references to support this. In addition, the authors could also project the current number using some kind of available growth parameters for relevant variables?

Thank you for this comment. We added a citation on line 174, page 8 as the reviewer suggested to show that the number of staffed beds, which is a proxy for the number of staff, has only grown by 2% from 2015-2021. Further, our model was created with the hope that future users could take their current data to perform the same calculations, which is why it outside of our aims to project growth of relevant variables. 

5. I found it a bit difficult in following step 1 and 2 where there were regressions performed. I may have missed this, but how do exactly the coefficients obtained from the regressions were used in the overall calculation? Please elaborate.

Thank you for this comment. We described this process in Table 3 Action 1A, but for further clarification, we added a paragraph with an example of how the regression model was used to estimate the number of staff on page 15, line 228- line 240, page 16.

Results

1. Page 19 line 303 – 304: I believe Step 2 here refers to the training, correct? Just for precaution, this may be misleading as readers may be confused between step 2 of the formula or step 2 as the training itself, even though there is already an explanation of step 2 of the BFHI (readers may have missed this).

We agree that this was confusing so we changed the “Steps” in Table 3 to “Actions” and added a line to clarify this language on page 10, line 201.

2. I may have missed this, but were the costs between US and Mexico compared using PPP?

We thank the reviewer for this comment. As such, we converted all Mexico costs to PPP in USD to make them comparable between the countries. We added our PPP conversion methodology on line 300, page 18. We presented new PPP cost in Table 5, Table 7, and Table 8. This largely did not change our results, but we adjusted the numbers and text where applicable in the abstract, on page 23, 24 and 26.

Discussion

1. Page 24 Line 392: I am not sure it is the place of the study to state that BHFI training had poor cost-effectiveness in small hospitals with low obstetric care specialization. Please consider whether the discussion should be on lower cost efficiency instead, which relates with the following arguments in the paragraph.

We agree. We changed line 417 on page 25 that said small hospitals have “poor BFHI training cost effectiveness” to “future…analyses… may show…’lower cost effectiveness’”. 

2. Page 26 line 433: this is important, please add references to support the statement.

We added the missing citation [5] for this claim on page 25.

3. Page 26 line 431. I agree that the future studies on cost effectiveness should be conducted, but I’m not sure whether the paragraph should stay in that context. The paragraph discusses on the more efficient approach of the training, which directly relates with the study. The need for cost effectiveness studies would be better placed at the end of the paragraph.

Thank you for this suggestion. We moved the sentence on comparing the cost effectiveness of training modalities to the end of the paragraph on page 27.

4. Page 27, limitations: I think the authors skipped the third limitation?

Thank you for catching this error. We corrected it.

5. Limitations should also consider the possibility of differences in local characteristics which cannot be fully captured by the current method. This is actually the strength and weakness of micro costing which the authors mentioned in the background. However, it is also the weakness of the current method which can be used more generally, but may have missed local context variations.

Thank you for this succinct summary of the high-level methodological strengths and limitations. We added this to the limitations section on page 29, line 502.

References

1. Brown HS. Physician demand for leisure: Implications for cesarean section rates. J Health Econ. 1996;15: 233–242. doi:10.1016/0167-6296(95)00039-9

2. Instituto Nacional de Estadística y Geografía (INEGI) [National Institute of Statistics and Geography]. Natalidad: Datos abiertos [Natality: Open data]. 

3. Balogun OO, Dagvadorj A, Yourkavitch J, Da Silva Lopes K, Suto M, Takemoto Y, et al. Health Facility Staff Training for Improving Breastfeeding Outcome: A Systematic Review for Step 2 of the Baby-Friendly Hospital Initiative. Breastfeed Med. 2017;12: 537–546. doi:10.1089/bfm.2017.0040

4. Dellifraine J, Langabeer J, Delgado R, Williams JF, Gong A. A transition strategy for becoming a Baby-Friendly Hospital: Exploring the costs, benefits, and challenges. Breastfeed Med. 2013;8: 170–175. doi:10.1089/bfm.2012.0050

5. OECD. Health at a Glance 2021: OECD Indicators. Paris; 2021. doi:10.1787/ae3016b9-en

---

## [Decision Letter · Decision Letter 1]

4 Aug 2022

How much does it cost to implement the Baby-Friendly Hospital Initiative training step in the United States and Mexico?

PONE-D-21-30699R1

Dear Dr. Arslanian,

We’re pleased to inform you that your manuscript has been judged scientifically suitable for publication and will be formally accepted for publication once it meets all outstanding technical requirements.

Kind regards,

Susan Horton

Academic Editor

PLOS ONE

Additional Editor Comments (optional):

Reviewers' comments:

Reviewer's Responses to Questions

**Comments to the Author**

1. If the authors have adequately addressed your comments raised in a previous round of review and you feel that this manuscript is now acceptable for publication, you may indicate that here to bypass the “Comments to the Author” section, enter your conflict of interest statement in the “Confidential to Editor” section, and submit your "Accept" recommendation.

Reviewer #1: All comments have been addressed

Reviewer #2: All comments have been addressed

2. Is the manuscript technically sound, and do the data support the conclusions?

Reviewer #1: Yes

Reviewer #2: Yes

3. Has the statistical analysis been performed appropriately and rigorously? 

Reviewer #1: Yes

Reviewer #2: Yes

4. Have the authors made all data underlying the findings in their manuscript fully available?

Reviewer #1: Yes

Reviewer #2: Yes

5. Is the manuscript presented in an intelligible fashion and written in standard English?

Reviewer #1: Yes

Reviewer #2: Yes

6. Review Comments to the Author

Reviewer #1: The authors dud a fantastic job revising the manuscript and addressing all comments. I believe this manuscript is ready and should be accepted for publication. I do have one very minor phrase addition for final publication that would be helpful though is not mandatory - page 27 line 262 after Consumer Price Index add for all goods and services. This clarifies that the authors did not use medical inflation of the CPI which is often much higher and saves the reader from having to check the reference to verify which inflation calculator was used. Thank you to authors for this excellent work.

Reviewer #2: Authors have addressed all of my comments thoroughly. I have no further suggestion and I believe it is now publishable.

7. PLOS authors have the option to publish the peer review history of their article (what does this mean?). If published, this will include your full peer review and any attached files.

Reviewer #1: No

Reviewer #2: No

---

## [Editor Report · Acceptance letter]

19 Sep 2022

PONE-D-21-30699R1 

How much does it cost to implement the Baby-Friendly Hospital Initiative training step in the United States and Mexico? 

Dear Dr. Arslanian:

I'm pleased to inform you that your manuscript has been deemed suitable for publication in PLOS ONE. Congratulations! Your manuscript is now with our production department. 

Kind regards, 

on behalf of

Dr. Susan Horton 

Academic Editor

PLOS ONE